# Exploring the Organic Acid Secretion Pathway and Potassium Solubilization Ability of *Pantoea vagans* ZHS-1 for Enhanced Rice Growth

**DOI:** 10.3390/plants13141945

**Published:** 2024-07-15

**Authors:** Shiqi Tian, Yufeng Xu, Yanglin Zhong, Yaru Qiao, Dongchao Wang, Lei Wu, Xue Yang, Meiying Yang, Zhihai Wu

**Affiliations:** 1College of Life Sciences, Jilin Agricultural University, Changchun 130118, China; 20220822@mails.jlau.edu.cn (S.T.); 20220839@mails.jlau.edu.cn (Y.X.); 17870145051@163.com (Y.Z.); 13315128845@163.com (Y.Q.); lwu@jlau.edu.cn (L.W.); xyang@jlau.edu.cn (X.Y.); 2Faculty of Agronomy, Jilin Agricultural University, Changchun 130118, China; wangdongchao9021@163.com

**Keywords:** potassium solubilization, *Pantoea vagans*, rhizosphere growth promoting bacteria, organic acid metabolism, microbial community

## Abstract

Soil potassium deficiency is a common issue limiting agricultural productivity. Potassium-solubilizing bacteria (KSB) show significant potential in mitigating soil potassium deficiency, improving soil quality, and enhancing plant growth. However, different KSB strains exhibit diverse solubilization mechanisms, environmental adaptability, and growth-promoting abilities. In this study, we isolated a multifunctional KSB strain ZHS-1, which also has phosphate-solubilizing and IAA-producing capabilities. 16S rDNA sequencing identified it as *Pantoea vagans*. Scanning electron microscopy (SEM) showed that strain ZHS-1 severely corroded the smooth, compact surface of potassium feldspar into a rough and loose state. The potassium solubilization reached 20.3 mg/L under conditions where maltose was the carbon source, sodium nitrate was the nitrogen source, and the pH was 7. Organic acid metabolism profiling revealed that strain ZHS-1 primarily utilized the EMP-TCA cycle, supplemented by pathways involving pantothenic acid, glyoxylic acid, and dicarboxylic acids, to produce large amounts of organic acids and energy. This solubilization was achieved through direct solubilization mechanisms. The strain also secreted IAA through a tryptophan-dependent metabolic pathway. When strain ZHS-1 was inoculated into the rhizosphere of rice, it demonstrated significant growth-promoting effects. The rice plants exhibited improved growth and root development, with increased accumulation of potassium and phosphorus. The levels of available phosphorus and potassium in the rhizosphere soil also increased significantly. Additionally, we observed a decrease in the relative abundance of *Actinobacteria* and *Proteobacteria* in the rice rhizosphere soil, while the relative abundance of genera associated with acid production and potassium solubilization, such as *Gemmatimonadota*, *Acidobacteria*, and *Chloroflexi*, as well as *Cyanobacteria*, which are beneficial to plant growth, increased. These findings contribute to a deeper understanding of the potassium solubilization mechanisms of strain ZHS-1 and highlight its potential as a plant growth-promoting rhizobacteria.

## 1. Introduction

Potassium plays a crucial role in plant metabolic activities, such as sugar metabolism and photosynthesis [1]. Additionally, potassium can regulate transpiration, affecting plants’ responses to cold, heat, or oxidative stress [2]. More than 90% of potassium in nature exists in the form of insoluble rocks and silicate minerals [3], serving as natural raw materials for potassium fertilizer production. According to estimates released by the U.S. Geological Survey [4], potassium ore is expected to last about 400 years at the current mining rate, indicating that current reserves are sufficient. However, when potassium fertilizer is applied to the soil, it is easily fixed by the soil, becoming ineffective or only slowly available to plants. This makes it difficult for plants to directly absorb and utilize the potassium, leading to a decreased utilization rate of potassium fertilizer and a waste of resources, which is detrimental to the sustainable development of agriculture [5].

Potassium-solubilizing bacteria (KSB), also known as silicate bacteria, are a class of bacteria isolated from soil and plant rhizospheres. They can dissolve aluminosilicate and apatite minerals, converting potassium in the soil from an insoluble to a soluble form that can be absorbed by plants, thereby increasing the content of available potassium in the soil [6]. KSB can enhance plant biomass by improving the soil’s physical and chemical properties and by secreting hormones that promote plant growth [7]. Therefore, introducing microbial activation methods to enhance fertilizer utilization is an effective way to meet plant growth and reduce the use of chemical fertilizers [8]. Bacteria such as *Bacillus* [9], *Burkholderia* [10] and *Agrobacterium* [11] have demonstrated potassium-solubilizing potential. Saha et al. [12] reported that a strain of *Bacillus licheniformis* released 7.22 mg/L of potassium from biotite after 21 days of inoculation. Muthuraja et al. [9] reported that *Burkholderia cenocepacia* with K solubilization from mica is 1.34 mg/L after 96 days. *Agrobacterium tumefaciens*, isolated from north-central India, released 49.73 mg/L and 16.20 mg/L of potassium from biotite and muscovite, respectively, over 21 days [11].

Currently, the mechanism by which KSB dissociate insoluble potassium is mainly concentrated in the following four aspects: First, KSB can produce organic and inorganic acids to acidify the soil, thereby changing insoluble potassium into soluble potassium and increasing the available potassium in the soil [13]. Chen et al. [14] studied the potassium solubilization mechanism of *Bacillus aryabhattai* SK1-7. They found that it had the highest organic acid yield under the condition of an insoluble potassium source, reaching 183.14 ng/mL at 168 h. The primary acids detected were formic acid, citric acid, acetic acid, and gluconate, with smaller amounts of fumaric acid and oxalic acid. Muthuraja Raji et al. [8] reported that four potassium-solubilizing bacteria—*Bacillus subtilis*, *Bacillus cereus*, *Bacillus licheni-formis*, and *Burkholderia* produced varying amounts of acetic acid and benzoic acid, selectively secreting ascorbic acid, malic acid, and oxalic acid to achieve potassium solubilization. Secondly, regarding the role of plant hormones, a variety of KSB can produce beneficial regulatory substances; these substances can effectively promote the growth and development of plants and can effectively use potassium [15]. The mechanism of potassium solution is explained by improving the absorption and utilization efficiency of potassium in plants. Thirdly, KSB can secrete various extracellular enzymes [16], including proteases, amylases, and cellulases, which decompose organic matter and release significant amounts of potassium. Additionally, Hu et al. [17] found that *Bacillus subtilis*, *Clostridium*, and *Thiobacillus* can produce exopolysaccharides with strong biodegradability for potassium feldspar and illite, thus releasing potassium ions. In conclusion, the potassium solubilization mechanisms vary among different strains, and the same strain may exhibit multifunctional characteristics. Therefore, exploring these mechanisms and fully understanding the processes involved in KSB activity will help promote the development of KSB microbial bactericides and their application in agriculture, contributing to the sustainable development of green agriculture.

*Pantoea* has been reported to have the functions of a heavy metal chelating agent [18] and soluble potassium solubilization [19]. Pot experiments showed the characteristics of dissolving phosphate, inhibiting the growth of pathogenic bacteria and improving plant growth and yield [20]. Therefore, it is increasingly considered as an ideal candidate for plant growth-promoting rhizobacteria [21]. However, different KSB strains exhibit diverse solubilization mechanisms, environmental adaptability, and growth-promoting abilities. In this study, strain ZHS-1, capable of solubilizing potassium and phosphate and secreting IAA, was isolated and identified as *Pantoea vagans* using 16S rDNA analysis. LC-MS/MS targeted organic acid detection was utilized to determine the types of organic acids secreted by the strain during potassium dissolution and their metabolic pathways via KEGG. Based on the identification of the metabolic network of various active substances produced by the strain, the growth-promoting effect of the strain through potassium solubilization and IAA secretion was verified by pot experiments. This study holds significant value in revealing the growth-promoting effect of this strain on plants.

## 2. Results

### 2.1. Screening and Identification of KSB

A strain of KSB with better potassium-dissolving ability, named ZHS-1, was screened according to the potassium-dissolving zone. The amount of potassium released by strain ZHS-1, as shown in Figure 1a, increased rapidly on the first day. In addition, the amount of potassium released on days 4–6 was significantly higher than that of the first 3 days, slightly decreasing to 18.4 mg/L after 7 days. As the potassium content increased, the pH value of the solution rapidly decreased on the first day and then stabilized, with the pH of the culture medium remaining between 3.37 and 3.43 within 7 days. Strain ZHS-1 was inoculated into a phosphate-solubilizing medium and cultured for 7 days. The soluble phosphorus in the supernatant peaked at 276 mg/L on the 3rd day, with the pH value dropping to 4.34 on the first day and then increasing to 5.37, where it stabilized (Figure 1b). The decrease in the pH value of the medium initially indicated that strain ZHS-1 produced a certain number of acidic substances during the dissolution of potassium and phosphate. The amount of acid produced during the potassium dissolution process was significantly higher than during the phosphate dissolution process. Strain ZHS-1 was cultured for 12 h to determine IAA yield, resulting in 8.14 mg/L, which was substantially different from distilled water and was used instead of bacterial supernatant as the negative control (CK1). (Figure 1c). Changes in potassium feldspar before and after exposure to the strain were observed using scanning electron microscopy (SEM; Hitachi, S-4800, Tokyo, Japan). As shown in Figure 1d, the surface of the untreated potassium feldspar was smooth and angular (Figure 1d). The surface was eroded after co-cultivation with the strain (Figure 1e). The morphology of the ZHS-1 strain was elliptical rod-shaped (Figure 1f). Homology comparison analysis was performed on the NCBI database using BLAST, and a phylogenetic tree was constructed using the neighbor-joining method of MEGA 7 software (Figure 1g). The results showed that the strain ZHS-1 was closely related to the *Pantoea vagans OsEp* strain, and the sequence similarity was 99%. Based on this, the ZHS-1 strain was identified as *Pantoea vagans*.

### 2.2. Optimization of Potassium-Solubilizing Conditions of Strain ZHS-1

According to the single-factor test, strain ZHS-1 achieves the highest potassium solubilization at pH = 6. The maximum potassium solubilization using glucose as the carbon source is 19.2 mg/L (Figure 2b), and with sodium nitrate as the nitrogen source, it is 19.03 mg/L (Figure 2c). Based on the results of the single-factor test, the three factors—carbon source, nitrogen source, and pH—were chosen as independent variables, with the potassium solubilization amount as the response value for the experimental design. Design-Expert 13.0 software was used to analyze the results, as depicted in Figure 2d–f. The optimal conditions for potassium solubilization by ZHS-1 were determined as follows: maltose and sodium nitrate were used as the carbon and nitrogen sources, respectively, with a pH of 7, resulting in a potassium solubilization amount of 20.7 mg/L. The optimal response surface conditions determined by software analysis were verified, with the potassium solubilization amount measured at 20.3 mg/L, closely aligning with the model’s predicted value.

### 2.3. Changes in Organic Acid Content and Key Enzyme Activity in the Process of Potassium

The amount of potassium released by strain ZHS-1 was closely related to changes in the pH value of the culture medium. Additionally, the pH of strain ZHS-1 exhibited significant changes within 24 h of growth in the potassium-solubilizing medium, particularly noticeable between 2 and 4 h, with the pH stabilizing at around 8 h (Appendix A). Therefore, three-time points—2 h, 4 h, and 8 h—were chosen to analyze the metabolic spectrum of organic acids produced by strain ZHS-1. Figure 3a illustrates the excellent repeatability among the treated samples, with significant differences observed between the treatments. A total of 65 metabolites were identified across the three time periods, with 15 organic acids exhibiting substantial changes in content at each time point (Appendix A). As shown in the radar chart depicting changes in organic acids, the content of methylsuccinic acid, pyruvic acid, adipic acid, and malic acid decreased from 2 h to 4 h (Figure 3b1). Conversely, the contents of IAA, maslinic acid and α-ketoglutaric acid significantly increased, with indole acetic acid exhibiting the largest fold difference of 1.604. Comparing the contents at 2 h and 8 h (Figure 3b2), it can be observed that the levels of m-methylsuccinic acid, pyroglutamic acid, and fumaric acid decreased continuously, whereas the contents of 2-hydroxyphenylacetic acid, α-ketoglutaric acid, IAA, cis-aconitic acid, azelaic acid, and 3-phenyllactic acid increased. Among these, α-ketoglutaric acid showed the most significant fold difference. The content of methylsuccinic acid and fumaric acid decreased between 4 h and 8 h (Figure 3b3), whereas the content of α-ketoglutaric acid and IAA increased, with the fold difference being greater than zero. Over the 2- to 8-h period, pyruvate, pantothenic acid, malic acid, coumaric acid, ferulic acid, and chlorogenic acid were down-regulated, whereas lactic acid, tartaric acid, anthranilic acid, α-ketoglutaric acid, succinic acid, IAA, and cis-aconitic acid were up-regulated. KEGG analysis (Appendix A) indicated that these organic acids were mainly derived from the glycolytic pathway and tricarboxylic acid cycle, with certain organic acids also being distributed in pathways such as tryptophan metabolism, pantothenate and coenzyme A biosynthesis, butanoate metabolism, and glyoxylic acid and dicarboxylate metabolism.

Therefore, the changes in the activity of key enzymes in the EMP-TCA metabolic pathway were measured at 2, 4 and 8 h. During potassium release by strain ZHS-1, the activity of citrate synthase increased rapidly from 2 h to 4 h, reaching an enzyme activity of 0.234 U/mL at 8 h, with no significant change compared to that at 4 h (Figure 3c). Isocitrate dehydrogenase activity decreased from 3.29 U/10^4^ cells at 2 h to 1.71 U/10^4^ cells at 8 h (Figure 3d). The activity of α-ketoglutarate dehydrogenase peaked at 4 h, reaching 0.34 U/10^4^ cells (Figure 3e). The pattern of change in lactate dehydrogenase activity mirrored that of citrate synthase. The enzyme activity increased from 2 to 4 h, reaching 0.106 U/10^4^ cells at 4 h. However, it tended to stabilize from 4 to 8 h, with the enzyme activity remaining at 0.105 U/10^4^ cells at 8 h (Figure 3f).

### 2.4. Growth-Promoting Effect of ZHS-1 on Rice Plants

The phenotype of rice and the content of phosphorus and potassium in the plant and soil of each treatment were measured.

To detect the growth-promoting effect of strain ZHS-1 on rice, a pot experiment was conducted to study its impact on the growth and development of rice. The phenotype of rice seedlings is shown in Figure 4a1–a3 and compared to treatment with nitrogen fertilizer alone (CK1) (Figure 4a1), treatment with nitrogen fertilizer combined with potassium bacteria (T1) (Figure 4a3) exhibited vigorous growth and increased tillering, though the growth of T1 plants was slightly less robust than treatment with nitrogen, phosphorus, and potassium fertilizer (CK2). From the root scanning images (Figure 4a4–a6), it is evident that the rice roots of T1 had abundant root hairs. The root length increased by 30.07% compared to CK1, and the root dry weight increased by 75.09% (Appendix A), though the root indexes were slightly lower than CK2. The analysis of potassium accumulation in rice plants’ roots, stems, and leaves at the seedling stage, full heading stage, and mature stage are shown in Figure 4b1–b3. The variation of potassium accumulation in the roots, stems, leaves, and total potassium of T1 and CK2 at the seedling and mature stages was the same and was significantly higher than that of CK1. However, the potassium content in the roots and leaves of T1 was significantly lower than that of CK2. At the full heading stage, the potassium accumulation in the roots, stems, leaves and whole plants of T1 and CK2 remained similar to that at the seeding stage, being significantly higher than that of CK1. The difference between T1 and CK2 was significant only in the roots, with no significant difference in the aboveground parts. Figure 4c1–c3 show that phosphorus accumulation and total phosphorus accumulation in different parts of the plants at the seedling, full heading, and mature stages were also significantly higher in T1 and CK2 than in CK1. At the mature stage, phosphorus accumulation in each part of T1 was significantly lower than that of CK2. There was no significant difference in phosphorus accumulation in the stems at the seedling and full heading stages. Still, there were significant differences in the roots and leaves at the seedling stage and the leaves at the full heading stage. From Figure 4d1,d2, it can be seen that CK1 showed no significant change in soil available potassium and available phosphorus at each period. In contrast, the soil’s available potassium and available phosphorus of T1 and CK2 showed an increasing trend as the growth period advanced, which was significantly higher than CK1 in each period. Compared with CK1, the soil-available potassium content of T1 increased by 85.75%, and the soil-available phosphorus increased by 92.8%. However, the differences between T1 and CK2 were significant, with T1 being lower than CK2. The chemical properties of the planting soil are shown in Appendix A.

### 2.5. Changes in the Microbial Community in Rice Rhizosphere Soil

The diversity and composition of microbial communities in the rhizosphere soil of three samples were analyzed using high-throughput sequencing technology. The PCoA diagram of Bray–Curtis distance (Figure 5a) illustrated that multiple sample points of the same treatment clustered together and were distributed in different quadrants. This indicates significant differences in microbial diversity and richness among the three samples, primarily due to the different treatments. The Venn diagram (Figure 5b) of the OTUs showed that 1299 OTUs identified in this experiment were shared among the three groups. The T1 treatment group had the most OTUs, totaling 7608, indicating that the bacterial treatment group caused the greatest change in soil microbial community composition. As shown in Figure 5c–f, the Chao1, Observed-species, Shannon, and Simpson indexes of the T1 treatment were significantly higher than those of CK2, indicating that inoculation with the ZHS-1 strain enhanced soil microbial species richness and diversity. From the phylum-level bacterial taxa shown in Figure 5g, it can be seen that *Chloroflexi*, *Actinobacteria*, *Proteobacteria*, *Gemmatimonadota*, *Acidobacteria*, and *Cyanobacteria* are relatively abundant in the three treatments, accounting for more than 90% of the bacterial community. Compared with the other two groups, *Actinobacteria* and *Proteobacteria* in T1 showed a higher relative abundance.

## 3. Discussion

In this study, a potassium-solubilizing, phosphate-solubilizing, and IAA-producing *Pantoea* strain, ZHS-1, was isolated. The potassium-solubilizing capacity reached 18.4 mg/L, the phosphate-solubilizing capacity was 276 mg/L at 3 days, and 8.14 mg/L IAA was produced at 12 h. This aligns with previous findings by Maria Rasul et al. [22] and Bakhshandeh et al. [23], who reported that *Pantoea* can dissolve phosphate and produce IAA. Currently, most scholars believe that strains often produce organic or inorganic acids to acidify the soil, making the direct solubilization mechanism of insoluble potassium to soluble potassium more common [13]. Liu et al. [24] found that *Fibrobacter* and *Paenibacillus* produced tartaric acid, oxalic acid, malic acid, citric acid, lactic acid, and acetic acid, which directly affected the potassium solubilization efficiency of the strains. In this study, strain ZHS-1 significantly decreased the pH value of the medium during the potassium and phosphorus dissolution processes. It was preliminarily inferred that the strain might dissolve potassium by secreting acids. The results of targeted organic acid metabolism revealed that the ZHS-1 strain produced a total of 65 organic acids during the potassium-solubilizing process. The glycolytic pathway and tricarboxylic acid (TCA) cycle are the metabolic hubs producing precursors for biosynthesis and energy production. Raghavendra et al. [25] demonstrated that the potassium-solubilizing process of the strain required a lot of energy. In this study (Figure 6), the types and amounts of acids produced by the ZHS-1 strain in the glycolytic pathway and TCA cycle were significantly higher than those in other metabolic pathways. This may indicate that glycolytic pathway and TCA cycle should be strengthened during the potassium release process to meet organic acid secretion and energy demands. The activities of citrate synthase, isocitrate dehydrogenase, and α-ketoglutarate dehydrogenase also confirmed this conclusion. Pantothenate is the precursor for coenzyme A synthesis, and glyoxylate and dicarboxylate metabolism are related to energy metabolism [26]. These three metabolic pathways may also ensure the solubilization of potassium and phosphate during the organic acid secretion and energy metabolism of strain ZHS-1. However, potassium-solubilizing bacteria activate insoluble potassium by producing organic acids, thereby reducing the environmental pH (Appendix A). Excessive acidification is not conducive to the strain’s growth. Fait A et al. [27] showed that the butanoate metabolism pathway, as a bypass of the TCA cycle, can regulate pH and acid-base balance and enhance adaptability in organisms. In this study, the organic acid metabolism spectrum of ZHS-1 also showed that the content of multiple intermediates in the butanoate metabolism process increased. This suggests that this process may play an important role in regulating the acid-base balance of environmental pH while participating in the potassium release of the strain. The IAA produced by microorganisms can regulate their physiological functions, adaptation to external stresses, and establishment of microbial interactions. Studies have shown that the IPDC pathway for converting tryptophan (Trp) to indole-3-acetic acid (IAA) is highly conserved at the gene and protein levels in *Pantoea* strains [28]. However, IAA was found to be synthesized through a tryptophan-independent pathway in a very small number of microorganisms [29]. In this study, the functional identification of strain ZHS-1 and the organic acid metabolism spectrum revealed that the strain could produce IAA, and its content increased with time within 2–4 h of potassium release, as indicated by the organic acid metabolism spectrum. In summary, the ZHS-1 strain mainly produces a large number of organic acids and energy through EMP-TCA, supplemented by pantothenic acid, glyoxylic acid, dicarboxylate metabolism, and other metabolic processes, thereby realizing the solubilization function of potassium through the direct solubilization mechanism. The solubilization of potassium and phosphorus by the strain is not a single metabolic process but rather the result of the interaction between strain growth and reproduction, nutrient supply, and energy metabolism.

Potassium and phosphorus are vital macronutrients necessary for plant development, yet they are often present in insoluble forms. The deficiency of available phosphorus and potassium in soil severely constrains plant growth and yield [30,31]. Previous studies have demonstrated that the introduction of plant growth-promoting bacteria can activate soil elements and enhance soil fertility. For instance, inoculating soil with *Enterobacter hormoechei* notably increased soil potassium content and stimulated wheat root length growth [32]. Similarly, Zhao et al. [33] applied two isolated potassium-solubilizing bacteria, *Bacillus megaterium* and *Bacillus mucilaginosus*, to calcareous soil, resulting in increased available potassium content and enhanced pepper growth. Furthermore, Seoud et al. [34] co-inoculated phosphate-solubilizing bacteria and potassium-solubilizing bacteria in soil, leading to increased phosphorus and potassium availability, thus facilitating maize plant absorption of these elements. Potassium is particularly crucial for rice growth and development, with rice exhibiting the highest potassium demand among all crops [35]. In this study, the ZHS-1 strain was inoculated into the rhizosphere of rice, revealing significant growth-promoting effects on rice plants at various growth stages, including the seedling, full heading, and mature stages. Plants treated with the bacteria exhibited robust aboveground growth and notably improved root hair development compared to untreated plants. Additionally, there was a notable increase in potassium and phosphorus accumulation in plants, along with elevated levels of available phosphorus and potassium in the rhizosphere soil.

Soil nutrition not only impacts plant growth but also influences the species, composition, abundance, and activity of soil microorganisms. The pH level is crucial for the absorption and utilization of plant nutrients. In acidic soil, the high concentration of hydrogen ions affects the solubility and availability of calcium, magnesium, phosphorus, and other elements, making it difficult for plants to absorb and utilize these nutrients. Therefore, proper adjustment of soil pH can enhance nutrient absorption and utilization by plants, promoting their growth and development [36]. The application of potassium fertilizer, for example, can rapidly increase the abundance of microbial species that require substantial nutrients while reducing the abundance of those that do not [37]. Meng et al. [38] discovered that the soil microbial functional activity in the group treated with complete potassium fertilizer was significantly higher than in the under-fertilized group, with the former notably enhancing the complexity of the microbial community’s functional network structure. Similarly, Huang et al. [39] confirmed that the abundance, community composition, and functional diversity of rhizosphere microorganisms could be enhanced through the application of microbial agents. Soil pH was positively correlated with the actinobacterial genera [40]. Similarly, *Actinobacteria* abundance decreased with lower pH and increased at higher pH [41]. This study observed a decrease in the relative abundance of *Actinobacteria* and *Proteobacteria* in the rhizosphere soil of rice treated with ZHS-1, along with an increase in the relative abundance of *Gemmatimonadota*, *Acidobacteria*, and *Chloroflexi*. *Acidobacteria*, known for their acidophilic nature and which are commonly found in soil and sediments.

Spieck et al. [42] reported that *Chloroflexi* contains thermomicrobes and photosynthetic bacteria capable of growing and reproducing in high-temperature environments using oxygen. They can also use light for photosynthesis in anaerobic environments. This indirectly suggests that ZHS-1 increases the content of organic acids in the rhizosphere after inoculation of the rice rhizosphere, which is beneficial for the growth of *Acidobacteria*, while promoting the proliferation of *Chloroflexi*. This synergistic effect contributes to the solubilization of potassium and phosphate. *Cyanobacteria* offer advantages such as enhancing soil fertility, improving crop productivity, and reducing chemical fertilizer pollution. They can increase the diversity and activity of soil microorganisms through symbiosis or association with other microorganisms [43]. Xu et al. [44] demonstrated that suitable nitrogen and phosphorus concentrations benefit Cyanobacteria growth, but excessive or insufficient nitrogen, phosphorus, and potassium fertilization adversely affects them. In this study, it was observed that the relative abundance of *Cyanobacteria* significantly increased in CK2 treatment, where ZHS-1 bacteria were inoculated, and nitrogen, phosphorus, and potassium fertilizers were applied, compared to CK1. This could be attributed to the effective regulation of soil nitrogen, phosphorus, and potassium by ZHS-1, promoting the growth of *Cyanobacteria.*

## 4. Materials and Methods

### 4.1. Soil Sample Collection

Soil samples were collected from Zhenlai County, Baicheng City, and Jilin Province (122°88′ E, 45°40′ N) in September 2022. Surface soil samples were collected using a sterile knife, then transferred to autoclaved, plastic-sealed containers and stored in darkness at 4 °C. Subsequent testing was conducted within one week after collection.

### 4.2. Isolation and Identification of KSB

Ten grams of soil samples were weighed and diluted to 100 mL with sterile distilled water. After shaking for 30 min, the liquid was taken for double dilution. A 100 μL dilution was coated with potassium-solubilizing solid medium (1L medium containing 5 g glucose, 0.5 g yeast extract, 0.3 g MgSO_4_·7H_2_O, 2 g Na_2_HPO_4_, 0.5 g (NH_4_)_2_SO_4_, 0.03 g MnSO_4_·7H_2_O, 0.03 g FeSO_4_·7H_2_O, 15 g agar, and 2.0 g potassium feldspar powder as the insoluble potassium source, pH 7.2). The medium was autoclaved at 121 °C for 20 min, and 0.25% bromothymol blue dye was added to the medium after slight cooling. The inoculated culture dishes were sealed and cultured in an incubator at 30 °C for 48 h. The potassium solubilization ability of the strain was qualitatively evaluated based on the formation of a transparent zone and the color change of bromothymol blue dye from greenish blue to yellow. Single colonies with an excellent potassium-dissolving effect were selected for separation, purification, and subculture and then stored at −80 °C. Single colonies were picked and inoculated into 10 mL of LB, cultured at 30 °C and 150 rpm for 12 h, and 1 mL of bacterial solution was centrifuged and dried into powder. Then, 1 mL of bacterial solution was inoculated into the potassium-solubilizing liquid medium and cultured at 30 °C and 150 rpm for 7 days. Moreover, the potassium feldspar powder was taken out and dried into powder for later use, which was compared with the dried potassium feldspar powder without bacterial treatment. After that, it was fixed with 2% glutaraldehyde, rinsed with 0.1 M PBS buffer 3 times, dehydrated with an ethanol gradient, dried with a CO_2_ critical point dryer, and then loaded into an ion sputtering device for gold plating. Scanning electron microscopy (SEM; Hitachi, S-4800, Tokyo, Japan) was used to capture the bacterial image of ZHS-1 at a 20 kV accelerating voltage and a 1000-fold magnification. Based on the bacterial 16S rDNA sequence, sequence amplification and sequencing were performed using the following universal primers: forward 27F (AGAGTTTGATCATGGCTCAG) and reverse 1492R (ACGGTTACCTTGTTACGACTT). BLAST analysis was conducted using the NCBI nucleotide homology sequence database. Strains with high homology to the strain and belonging to different species were selected.

### 4.3. Determination of the Multifunctional Ability of KSB

#### 4.3.1. Determination of Potassium Dissolving Capacity

The single colony of ZHS-1 was inoculated into 10 mL of LB medium and cultured at 30 °C and 150 rpm for 12 h as seed liquid. Then, 1% of this seed liquid was added to 100 mL of potassium solution liquid medium and cultured at 30 °C and 150 rpm for 7 days. A potassium solution medium without the seed solution served as the blank control, with each treatment repeated 3 times. A total of 2 mL of culture solution was collected from each sample every day for 1 week. The samples were centrifuged at 8000 rpm for 10 min, and the supernatant was used to determine the pH value of the medium. The quantitative determination and calculation of soluble potassium were performed by a flame photometer (F-100, F-2211004, Shanghai, China).

#### 4.3.2. Determination of Phosphate-Solubilizing Capacity

The phosphate-solubilizing ability of the strain was evaluated using a liquid NBRIP medium containing 10 g of Ca_3_(PO_4_)_2_ as the sole phosphorus source. The preparation of the seed liquid was the same as described in Section 4.3.1. A total of 1 mL of seed liquid was inoculated into 100 mL of NBRIP liquid medium and cultured at 30 °C and 150 rpm for 7 days, with each treatment repeated 3 times. One milliliter of each sample was collected daily and centrifuged at 10,000 rpm for 5 min. The supernatant was then used to determine the pH value of the medium, and the change in soluble phosphorus content was determined using the molybdenum-antimony colorimetric method. An uninoculated NBRIP medium was used as a control.

#### 4.3.3. Determination of Indoleacetic Acid Yield

The production of indole acetic acid by the strain was determined according to Sachdev’s method. Following the seed solution culture method described in Section 4.3.1, a 12 h culture solution of ZHS-1 bacteria was obtained, centrifuged at 11,000 rpm for 15 min, and 1 mL of supernatant was added to 2 mL of Salkowski reagent. After incubation for 30 min, the appearance of a pink color in ZHS-1 indicated the presence of IAA. Distilled water was used instead of bacterial supernatant as the negative control (CK1), and 50 μg/mL IAA was used instead of bacterial supernatant as the positive control (CK2). The optimal density (OD) values of the three solutions were read at 530 nm to calculate the yield of IAA.

### 4.4. Response Surface Optimization of Potassium-Solubilizing Bacteria

The single-factor test of potassium-solubilizing conditions of strain ZHS-1 was conducted as follows: in the medium with different pH values (3, 4, 5, 6, 7, 8, 9, and 10), different carbon sources (glucose, sucrose, maltose, and mannitol) as the sole carbon source, and different nitrogen sources (ammonium sulfate, urea, ammonium chloride, and sodium nitrate) as the sole nitrogen source, cultures were maintained at 30 °C and 150 rpm for continuous culture for 3 days. The growth of the strain was determined, and the culture medium was centrifuged at 8000 rpm for 10 min. The supernatant was then taken to determine the potassium content. The colony counting method was used to calculate the growth of strain ZHS-1 at each pH. The three factors of carbon source (A), nitrogen source (B), and pH (C) were used as independent variables, and the amount of potassium solution (Y) was used as the response value to design the experiment. Design-Expert 13.0 software was used for analysis.

### 4.5. Organic Acid Metabolomics Analysis

#### LC-MS/MS Detection of Organic Acids

The seed liquid was prepared according to the method described in Section 4.3.1 and then transferred to 100 mL of potassium-dissolving liquid medium with a 1% inoculation amount. The seed liquid was continuously cultured at 30 °C and 150 rpm. Samples were collected at 2, 4, and 8 h for LC-MS/MS analysis.

### 4.6. Determination of Key Enzyme Activity in the Process of Organic Acid Production

The seed liquid was prepared according to the method described in Section 4.3.1 and then transferred to 100 mL of potassium-dissolving liquid medium with a 1% inoculation amount. Five million bacterial cells were collected at 2, 4 and 8 h for the activities of citrate synthase, isocitrate dehydrogenase, α-ketoglutarate dehydrogenase, and lactate dehydrogenase were determined using a kit method (BoxBio, Beijing, China).

### 4.7. Study on the Growth-Promoting Effect of KSB on Rice

#### 4.7.1. Cultivation and Sampling of Rice PLANTS

According to the culture method outlined in Section 4.3.1, the KSB culture with an OD_600_ of 0.8 was centrifuged at 4 °C at 9600× *g* for 10 min, the cells were washed once with sterile water and resuspended with 2 mL of sterile water. The rice seeds used in this study were selected from the local main cultivar, “Changjing 616”. Soil samples were collected from the experimental field of Jilin Agricultural University (125°41′ E, 43°80′ N). After removing the surface residue of the experimental field, soil was collected from a depth of 0–20 cm. Opaque plastic buckets with an upper diameter of 30 cm, a lower diameter of 27 cm, and a height of 30 cm were chosen. Each bucket was filled with 5 kg of subsoil. The experiment was divided into three treatments: nitrogen fertilizer alone (CK1), nitrogen, phosphorus, and potassium fertilizer (CK2), and nitrogen fertilizer combined with potassium bacteria (T1). Each treatment was repeated five times. Nitrogen fertilizer consisted of urea (46% N), phosphate fertilizer was calcium superphosphate (12% P_2_O_5_), and potassium fertilizer was potassium dioxide (60% K_2_O). Each barrel received pure nitrogen at 2.14 g, P_2_O_5_ at 3.84 g, and K_2_O at 0.76 g, along with all disposable base fertilizers. Bacterial liquid was applied at a rate of 2 mL/hole. Three holes were drilled in each barrel, with 8 rice seeds placed in each hole. After air-drying and sieving, the seeds were covered with fine soil and irrigated once every other day after sufficient watering. Rice roots, stems, leaves, and rhizosphere soil were sampled at the seedling, full heading, and maturity stage. The rice plants were carefully removed from the soil, minimizing root damage as much as possible. Loose soil around the roots was removed, and the roots were rinsed with water and scanned using a rhizosphere scanner (MRS-9600TFU2L, Shanghai, China). Each rice tissue’s dry weight was determined by oven drying at 70 °C. The rhizosphere soil was scraped with a sterile knife and placed in a sterile tube, then dried and stored at 4 °C for soil available phosphorus, available potassium, and microbial analyses. Rice root, stem, and leaf tissues were dried and preserved for the analysis of total phosphorus and total potassium in plant tissues. The specific methods for analyzing total phosphorus and total potassium in plant tissues and available phosphorus and potassium in soil were referenced from soil and agricultural chemistry analysis. Total phosphorus and total potassium in plant tissues were calculated using the formula: g/plant; k concentration × DW of the plant.

#### 4.7.2. Sequencing of Soil Microbial Diversity

The total soil DNA of each soil sample (0.5 g) was extracted using the Omega Soil DNA Kit (D5635-02) from Omega Bio-Tek, based in Norcross, GA, USA. The V3–V4 region of the 16S rDNA gene was amplified using primers 338F (5′-ACCCTACGGGGCAG-3′) and 806R (5′-GGACTACHVGGGTWTCTAAT-3′). Bioinformatics analysis of the microbiome was conducted using QIIME2 2019.4 software. The UniFrac distance metric was used for beta diversity analysis, with principal coordinate analysis (PCoA) used for visual representation. The results of microbial composition and richness at the phylum level were obtained using AVS classification and taxonomic status identification.

## 5. Conclusions

In this study, a potassium-solubilizing bacterium, *Pantoea vagans* ZHS-1, was isolated, which possesses the ability to solubilize potassium, dissolve phosphate, and produce IAA. The potassium-solubilizing capacity of this strain reached 20.3 mg/L when using maltose as the carbon source and sodium nitrate as the nitrogen source at pH 7. SEM results also confirmed that strain ZHS-1 can corrode the surface of compact and smooth potassium feldspar, rendering it rough and loose. Organic acid metabolic profiling revealed that strain ZHS-1 primarily utilizes the EMP-TCA cycle, supplemented by pathways involving pantothenic acid, glyoxylic acid, and dicarboxylic acids, to produce a large number of organic acids and energy. This enables the bacterium to solubilize potassium and phosphate through direct dissolution mechanisms, whereas IAA secretion depends on the tryptophan metabolic pathway. Strain ZHS-1 demonstrated a significant growth-promoting effect on rice, enhancing the activation of phosphorus and potassium in the rhizosphere soil and markedly increasing the accumulation of potassium and phosphorus in the plants. Inoculation with strain ZHS-1 led to a decrease in the relative abundance of *Actinobacteria* and *Proteobacteria* in the rhizosphere soil while increasing the relative abundance of *Gemmatimonadota*, *Acidobacteria*, *Chloroflexi*, and *Cyanobacteria*, which are beneficial for plant growth.

## Figures and Tables

**Figure 1 plants-13-01945-f001:**
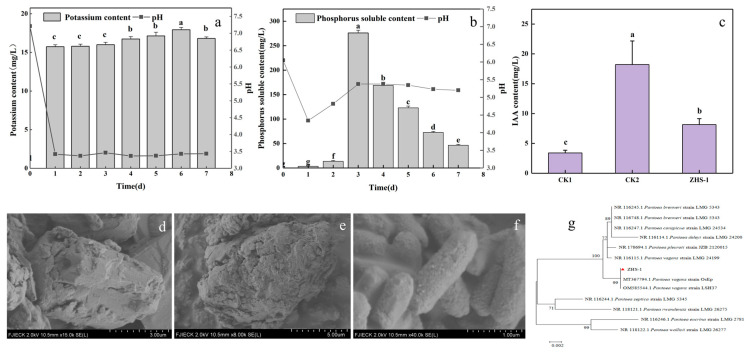
Isolation and identification of potassium-solubilizing bacteria ZHS-1. (**a**) Potassium content and pH change in 7 days. (**b**) Phosphate solubilizing content and pH change in 7 days. (**c**) IAA production. CK1: Distilled water was used instead of bacterial supernatant as the negative control. CK2: 50 μg/mL IAA was used instead of bacterial supernatant as the positive control. (**d**) SEM analysis of untreated potassium feldspar of strain. (**e**) SEM of potassium feldspar after strain treatment. (**f**) SEM of the strain. (**g**) Phylogenetic tree of the KSB strain based on 16S rDNA sequences. Different lowercase letters indicate significant differences (*p* < 0.05).

**Figure 2 plants-13-01945-f002:**
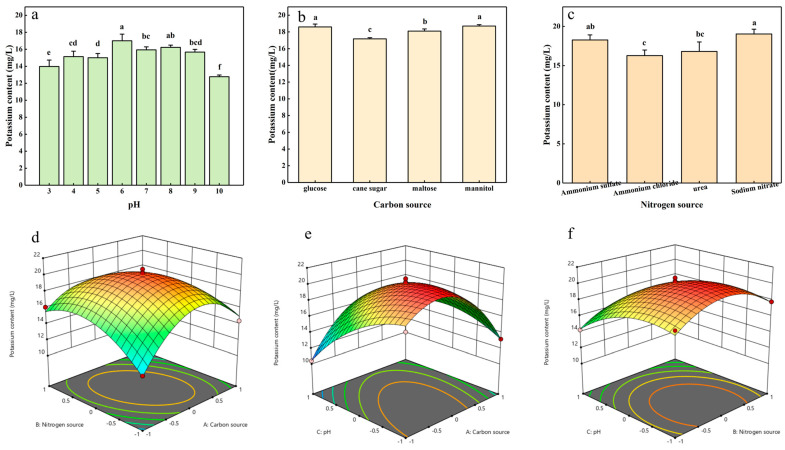
Analysis of potassium solubilization of strain ZHS-1. (**a**–**c**) Single factor tests in which pH, carbon source, and nitrogen source are varied. (**d**–**f**) Potassium release analyzed by response surface analysis of two-factor interactions between nitrogen source and carbon source, pH and carbon source, and pH and nitrogen source, respectively. Different lowercase letters indicate significant differences (*p* < 0.05).

**Figure 3 plants-13-01945-f003:**
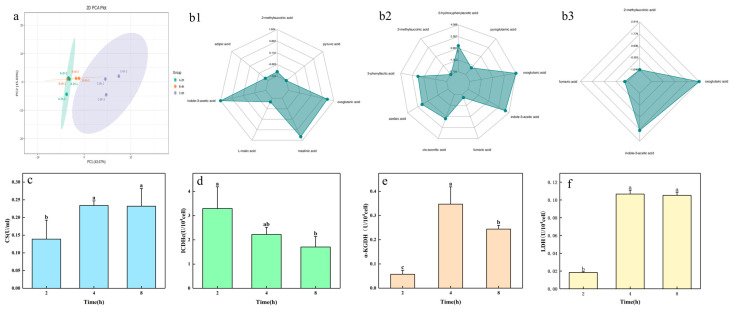
Changes in organic acid metabolism and key enzyme activities in the potassium-solubilizing culture medium of strain ZHS-1. (**a**) PCA of metabolites at 2 h, 4 h, and 8 h. (**b1**–**b3**) Radar charts of the first ten metabolites were performed at 2 and 4 h, 2 and 8 h, and 4 and 8 h, respectively. (**c**) Citrate synthase activity. (**d**) Isocitrate dehydrogenase activity, (**e**) α-ketoglutarate de-hydrogenase activity, (**f**) lactic dehydrogenase activity. Different lowercase letters indicate significant differences (*p* < 0.05).

**Figure 4 plants-13-01945-f004:**
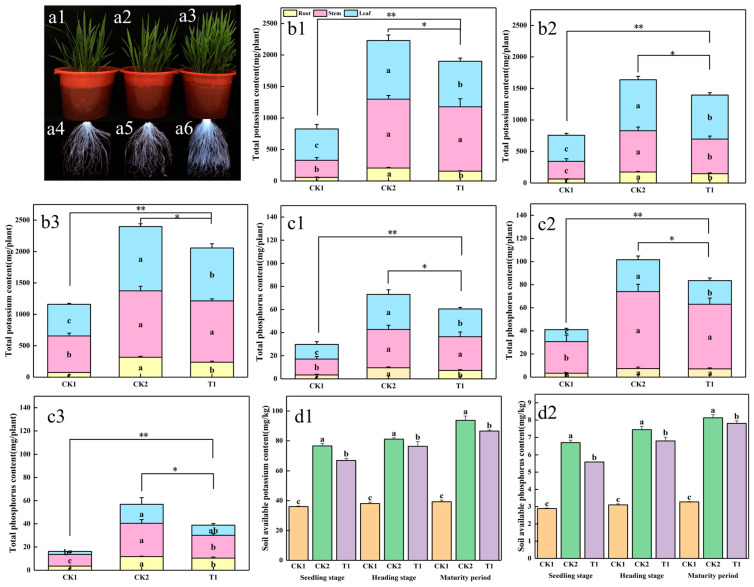
Effects of strains on phosphorus and potassium contents in rice seedlings during growth and development. (**a1**–**a6**) Plant phenotype and root scanning at the seedling stage. (**b1**–**b3**) The total potassium accumulation maps of rice roots, stems, and leaves at the seedling, full heading, and mature stage, respectively. (**c1**–**c3**) The total phosphorus accumulation maps of rice roots, stems, and leaves at the seedling, full heading, and mature stages, respectively. (**d1**,**d2**) Available potassium and available phosphorus of rhizosphere soil at the seedling, full heading, and mature stages. Different letters indicate significant differences (*p* < 0.05). * and ** indicate significant differences (*p* < 0.05).

**Figure 5 plants-13-01945-f005:**
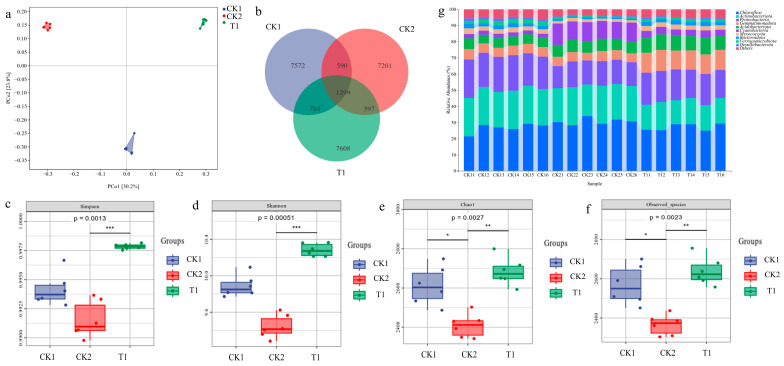
Microbial community diversity and composition of soils from different treatments. (**a**) Beta diversity reflected by the PCoA plot of Bray–Curtis distances between samples constrained by treatments. (**b**) Venn diagram of different samples for the exclusive and shared OTUs. (**c**–**f**) Alpha diversity is reflected by Chao1, observed species, Shannon, and Simpson indexes. *, ** and *** above the boxes indicate significant differences at *p* < 0.05 using Dunn’s test. (**g**) Bar plot of taxa composition at phylum level.

**Figure 6 plants-13-01945-f006:**
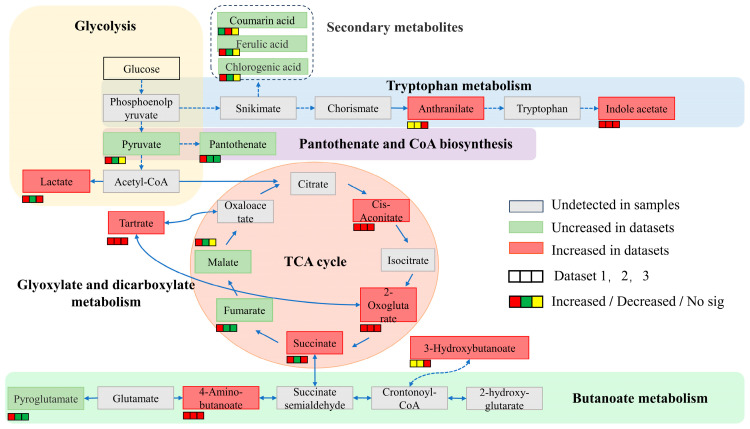
Organic acid metabolic profile of strain ZHS-1 during potassium solubilization.

## Data Availability

Data are contained within the article.

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
