# Peer review of "Exploring the Organic Acid Secretion Pathway and Potassium Solubilization Ability of *Pantoea vagans* ZHS-1 for Enhanced Rice Growth"

_plants, 2024, doi:10.3390/plants13141945_

Round 1
Reviewer 1 Report
Comments and Suggestions for Authors
The manuscript “Study on the Organic Acid Secretion Pathway and Potassium Solubilization Ability of Pantoea Vagans ZHS-1 in Promoting Effect on Rice Growth ” expounded the abilities of potassium solubilization, organic acid secretion of potassium-solubilizing bacteria Pantoea vagans ZHS-1 and its effects on the growth, phosphorus and potassium contents and rhizospheric soil microbial community structure of rice seedling. In general, this article is valuable for the research of plant growth promoting bacteria utilization. However, there are still lots of problems need to be improved before this article’s publication.
Author Response
|
1. Summary |
|
|
|
Thank you very much for taking the time to review this manuscript. Please find the detailed responses below and the corresponding revisions in track changes in the re-submitted files. |
||
|
2. Point-by-point response to Comments and Suggestions for Authors |
||
|
Comments 1: Introduction: Please pay attention to the correct writing format of Latin scientific names of the species, the authors should carefully check the entire text. And please write out the full name of A. tumefaciens (Line 56), although it is a common species, you still need to provide its full name at the first mention in the article. |
||
|
Response 1: According to your suggestion, we carefully corrected writing format throughout the entire text, such as Latin italics, capitalization, etc. We supplemented full name “Agrobacterium tumefaciens” in line 58 of the text.
|
||
|
Comments 2: Introduction: The introduction did not clearly tell the readers why you choose Pantoea vagans ZHS-1 as your research object, why it is valuable to study on. Please introduce this KSB, for example: the genus it belongs to, the former researches on it, its value, the problems and shortcomings in previous research on it, and what gaps can your research fill in the previous research... |
||
|
Response 2: The 10 strains with potassium solubilization ability were obtained during the initial screening, and tested their phosphate-solubilizing and IAA production abilities. Finally, we found that strain ZHS-1 had higher potassium solubilization ability than other strains, and also had phosphate solubilization and IAA production abilities. Therefore, we choose Pantoea vagans ZHS-1 as your research object. We have added an introduction about “the genus it belongs to, the former researches on it, its value”in line 86-90 of the introduction and supplemented literature [19]、[20]、[21] and [22]. 19. Ayansina Segun, A.; Olubukola Oluranti, B.; Oluwole Samuel, A. Bioflocculant production and heavy metal sorption by metal resistant bacterial isolates from gold mining soil. Chemosphere 2019, doi: 10.1016/j.chemosphere.2019.05.092. 20. Kour, D.; Rana, K.L.; Kaur, T.; Yadav, N.R.; Halder, S.K.; Yadav, A.N.; Sachan, S.G.; Saxena, A.K. Potassium solubilizing and mobilizing microbes: Biodiversity, mechanisms of solubilization, and biotechnological implication for alleviations of abiotic stress. 2020, 177–202, doi.org/10.3389/fmicb.2023.1196024 21. Luziatelli, F.; Ficca, A. G.; Cardarelli, M. T.; Melini, F.; Cavalieri, A.; and Ruzzi, M. Genome Sequencing of Pantoea agglomerans C1 Provides Insights into Molecular and Genetic Mechanisms of Plant Growth-Promotion and Tolerance to Heavy Metals. Microorganisms 2020, 8, doi:10.3390/microorganisms8020153. 22. Dutkiewicz, J., Mackiewicz, B., Lemieszek, M. K., Golec, M., and Milanowski, J. Pantoea agglomerans: a mysterious bacterium of evil and good. Part IV. Beneficial effects. Annals of agricultural and environmental medicine: AAEM 2016, 23, 206-222, doi:10.5604/12321966.1203879 Regarding “the problems and shortcomings in previous research on it, and what gaps can your research fill in the previous research”, we supplementted a brief summary in line 90-92.
Comments 3: Results-Optimization of potassium-solubilizing conditions of strain ZHS-1: In Figure 2b and Figure 2c, there was only one concentration for different carbon and nitrogen sources, and the author determined the optimal type of carbon and nitrogen source based on the results of one concentration, that was not rigorous. Why did not you set a concentration gradient for the different carbon and nitrogen sources during the optimization experiment? Response 3: We believe that this concentration is the best concentration in the potassium-solubilizing process. Therefore, the concentration gradient optimization experiment with different carbon and nitrogen sources was not selected. Thank you very much for your suggestion. We will conduct experiments on carbon and nitrogen source concentration gradients in the next step research to further optimize the potassium solubilization conditions of ZHS-1 strain. Alexander medium is a universal medium for screening potassium solubilizing bacteria. When optimizing the experimental conditions, we used the amount of carbon and nitrogen sources in this medium as the basis for setting the conditions. Therefore, the results presented in this article are only the optimization results at the same concentration.
Comments 4: Results-Phenotype of rice and content of phosphorus and potassium in the plant and soil of each treatment: Will you pleased to provide the original chemical properties of the planting soil if possible, such as pH, the contents of total N, total P, total K, available N, available P, available K, especially the contents of available N, available P and available K. Response 4: Thank you very much for your useful comments. We added the chemical properties of the planting soil according to your suggestion in Table S3 on page 15, such as total N, available P, available K and pH.
Comments 5: Changes in the microbial community in rice rhizosphere soil: This section is not suitable for placement under the previous heading “Growth-promoting effect of ZHS-1 on rice plants”. And I suggest you to kindly set it as a separate chapter. |
||
|
Response 5: Thank you very much for your comments. We have based on your comments to set“Changes in the microbial community in rice rhizosphere soil”as a separate chapter in the Results 2.5. See Line 248 for details.
Comments 6: Discussion: At the first paragraph of Discussion, I suggest the authors add some discussion on the impact of pH on soil environment, fertility status, microbial community structure, and plant growth. And the authors are also encouraged to focus on discussing the potential impact of species with significant changes in abundance in rhizopheric soil on soil environment and plant growth (eg. they may have been proven in previous studies that they can affect certain soil physico-chemical properties, or their metabolites can affect plant growth in certain ways, or they are potential plant pathogens...). Response 6: According to your suggestion, we have supplemented discussion on the impact of pH on soil environment and plant growth in the discussion in line 347-353 in the manuscript. Simultaneously adding literature [37]. In the discussion about the issue of “the potential impact of species with significant changes in abundance in rhizopheric soil on soil environment and plant growth”, we added the research results of them in the same field in line 361-353 in the revision, and added references [41] and [42]. 37. Naz, M.; Dai, ZC.; Hussain, S.; Tariq, M.; Danish, S.; Khan, IU.; Qi SS.; Du, D. The soil pH and heavy metals revealed their impact on soil microbial community. Journal of environmental management 2022, 321, 115770, doi: 10.1016/j.jenvman.2022.115770. 41. Zhalnina, K.; Dias, R.; de Quadros, P.D.; Davis-Richardson, A.; Camargo, F.A.; Clark, I.M.; McGrath, S.P.; Hirsch, P.R.; Triplett, E.W. Soil pH determines microbial diversity and composition in the park grass experiment. Microb. Ecol. 2015, 69, 395–406, doi:10.1007/s00248-014-0530-2. 42. Wang, C.; Zhou, X.; Guo, D.; Zhao, J.; Yan, L.; Feng, G.; Gao, Q.; Yu, H.; Zhao, L. Soil pH is the primary factor driving the distribution and function of microorganisms in farmland soils in northeastern China. Ann. Microbiol. 2019, 69, 1461–1473, doi: 10.1007/s13213-019-01529-9.
3. Response to Comments on the Quality of English Language |
||
|
Response: We have proofread the manuscript for English by a native English speaker. At the same time, we also carefully corrected the formatting of the entire text, such as Latin italics, uppercase, etc. |
||

Reviewer 2 Report
Comments and Suggestions for Authors
The paper is out of the scope of plants, as most of the reseacrh is performed in a bacterial strain, so I would advise the authors to resend it to a journal devoted to soil microbiology of plant-bacterial interactions.
Going into the data presented. Most of the experiments are presented in a very confuse way, and some results contradict what is observed in other figures. Many aspects need to be clarified and many figures improved.
Specifically:
Introduction: The description of the roles of potassium in plant physiology is naive, What does "physiological resistance" means? The main roles are not even mentioned. see for instance:
Modulation of potassium transport to increase abiotic stress tolerance in plants
JM Mulet, R Porcel, L Yenush Journal of Experimental Botany 74 (19), 5989-6005 Line 56: A. tumefacions is rhizobium radiobacter. Does it fall within the categories previously mentioned? Line 63: Authors say; "Currently, the mechanism by which potassium-solubilizing bacteria (KSB) dissociate insoluble potassium is mainly concentrated in the following four aspects:", but the second aspect is: "Secondly, many KSB produce plant hormones, which are beneficial regulatory substances that effectively promote plant growth and potassium utilization [15]." This has nothing to do with potassium solubilization. Please rewritte the whole paragraph. Figure 1: explain in the legend what is CK1 and CK2. Panel C. IAA is a plant hormone where it comes from when you are using mineral fertilizers? This is quite a nonsense. Panel G: nothing can be read. Figure 2: In panel a there is a discrepancy with figure 1a. In figure 1a the best ability to solubilize potassium is attained at pH 3,5, but here is at pH 6. Please, explain. Can the bacteria really growth at such different pHs? remarkable, Please include data on growth at each pH, as it is difficult to believe that the bacteria stands such a wide range. Conclusions: Line 500: Where have you determined the phosphate solubilization capacity of your strain? Also, where does the IAA comes from, as you have more IAA when you are using CK2 (figure 1) Table S1: Check the units as is difficult to believe that the root length is about 24 meters and the root surface is 6,4 m^2. Comments on the Quality of English LanguageNeeds to be revised thoroughly. The reading is quite difficult.
Author Response
|
1. Summary |
|
|
|
Thank you very much for taking the time to review this manuscript. Please find the detailed responses below and the corresponding revisions in track changes in the re-submitted files. |
||
|
2. Point-by-point response to Comments and Suggestions for Authors |
||
|
Comments 1: The paper is out of the scope of plants, as most of the reseacrh is performed in a bacterial strain, so I would advise the authors to resend it to a journal devoted to soil microbiology of plant-bacterial interactions. |
||
|
Response 1: We have contacted the editor before submitting the manuscript, and the editor thinks it is consistent with the scope “Interactions between Plants and Soil Microorganisms” of “Plant–Soil Interactions’’ in Plants. Therefor, we choose this journal.
|
||
|
Comments 2: Going into the data presented. Most of the experiments are presented in a very confuse way, and some results contradict what is observed in other figures. Many aspects need to be clarified and many figures improved. |
||
|
Response 2: We are sorry for our carelessness. Based on your comments, the entire text has been made detailed revisions and corrections, we confirmed all the contents of the charts according to the original experimental data, and the results showed that the data in the charts was correct. However, (1) the value of 298 mg/L dissolved phosphate in line 110 and line 271 of the text was wrong, so it was corrected to 276 mg/L according to Figure 1b. (2) The title of Table S1 in line 546 was incorrectly expressed, and it was corrected to "Phenotypic changes of rice seedlings under different treatments".
Comments 3: Introduction: The description of the roles of potassium in plant physiology is naive, what does "physiological resistance" means? The main roles are not even mentioned. Response 3: The term "physiological resistance" we want to express is that potassium not only participates in physiological processes such as plant sugar metabolism and photosynthesis, but also participates in plant resistance to abiotic stress. However, the text may not have been clearly expressed, which has caused you confusion. According to the description in the literature [2], we have revised this section to '' it can affect plant response to abiotic stress, such as cold, heat or oxidative stress’’ in line 37-38. 2. Mulet, JM.; Porcel, R.; Yenush, L. Modulation of potassium transport to increase abiotic stress tolerance in plants. J Exp Bot. 2023 Oct 13;74(19):5989-6005. doi: 10.1093/jxb/erad333.
Comments 4: Line 56: A. tumefacions is rhizobium radiobacter. Does it fall within the categories previously mentioned? Response 4: According to your question, we have added the classification of Agrobacterium tumefaciens in line 54 and readjusted the expression of the example of KSB in line 55 of the manuscript.
Comments 5: Line 63: Authors say; "Currently, the mechanism by which potassium-solubilizing bacteria (KSB) dissociate insoluble potassium is mainly concentrated in the following four aspects:", but the second aspect is: "Secondly, many KSB produce plant hormones, which are beneficial regulatory substances that effectively promote plant growth and potassium utilization [15]." This has nothing to do with potassium solubilization. Please rewritte the whole paragraph. |
||
|
Response 5: Potassium-solubilizing bacteria can use potassium in soil to provide nutrients for growth and development. The potassium-solubilizing mechanism mainly includes: It directly promotes the dissolution of soil potassium and improves the absorption and utilization efficiency of potassium by plants. In the manuscript, "Secondly, many KSB produce plant hormones, which are beneficial regulatory substances that effectively promote plant growth and potassium utilization [16]." It belongs to" the aspect of improves the absorption and utilization efficiency of potassium by plants to enhance the solubility of potassium."
Comments 6: Figure 1: explain in the legend what is CK1 and CK2. Panel C. IAA is a plant hormone where it comes from when you are using mineral fertilizers? This is quite a nonsense. Panel G: nothing can be read. Response 6: Panel C in Figure 1 is to identify the IAA production ability of strain ZHS-1. Therefore, CK1: distilled water was used instead of bacterial solution as the negative control, CK2 :50 μg/ml IAA was used instead of bacterial solution as the positive control. The IAA content of T1 is not from plants, but secreted by strain ZHS-1. At the same time, we have also provided supplementary explanations for the captions in Figure 1c at line 130-132. Panel G is phylogenetic tree of KSB based on 16S rDNA sequences. The purpose of this diagram is to identify the ZHS-1 strain. However, the description of the figure in the text may not be specific and clear enough. Therefore, in this revision, the results of Figure 1g were analyzed and supplemented in line 122-126 of the text.
Comments 7: Figure 2: In panel a there is a discrepancy with figure 1a. In figure 1a the best ability to solubilize potassium is attained at pH 3,5, but here is at pH 6. Please, explain. Can the bacteria really growth at such different pHs? remarkable, please include data on growth at each pH, as it is difficult to believe that the bacteria stands such a wide range. Response 7: The pH in Figure 1 is the real-time pH value of the culture medium that may be formed by the secretion of organic and inorganic acids during the potassium solubilization of strain ZHS-1. pH in Figure 2 refers to the initial pH values of 8 different gradient potassium solubilizing media prepared in order to obtain the optimal conditions of potassium solubilization for strain ZHS-1. In Figure 2, pH=6 is the most suitable potassium dissolution condition for this strain, which was obtained by single factor experiments We also noticed the issue of whether the strain can grow in the pH 3-10. Therefore, we used the colony counting method to calculate the growth of strain ZHS-1 at each pH, as shown in Figure S1 added in this revision. From Figure S1 in line 530, the strain ZHS-1 can grow well in the pH 4-8 medium, especially pH 6 and 7, and can hardly grow under the conditions of pH 3, 9, and 10. This may also be the reason why the potassium solubility cannot continue to increase when the pH of medium declined to around 3 in Figure 1a.
Comments 8: Conclusions: Line 500: Where have you determined the phosphate solubilization capacity of your strain? Also, where does the IAA comes from, as you have more IAA when you are using CK2 (figure 1) Response 8: Figure 1b in the text showed the results of phosphate solubilization capacity by ZHS-1 in 7 days. It can be seen from the figure that the maximum phosphate solubility is 276 mg/L on the third day. “Secreting IAA” in conclusion means that strain ZHS-1 has the ability to secrete IAA. The detailed explanation is the same as the sixth question above.
Comments 9: Table S1: Check the units as is difficult to believe that the root length is about 24 meters and the root surface is 6,4 m^2. Response 9: We have checked in detail the the units in Table S1 and the unit is correct. The result was obtained by the rhizosphere scanner (MRS-9600TFU2L, Shanghai), the root length represents the sum of all the root lengths of a rice plant, instead of the length of the root. The root surface area, root volume and root tip number were determined by the same method as that of root length.
3. Response to Comments on the Quality of English Language |
||
| Response: We proofread the English of the manuscript by native English speakers. At the same time, we also carefully corrected formatting throughout the entire text, such as Latin italics, capitalization, etc. | ||

Round 2
Reviewer 2 Report
Comments and Suggestions for Authors
Authors have made a great effort to improve the manuscript. I can recommend publication.
Author Response
Dear Reviewer:
First of all, we would like to take this opportunity to appreciate the time and efforts made by the reviewer. Your constructive comments could make our manuscript more readable and help clarify any potential confusions. At the same time, thank you very much for your recommending.
Best wishes,
Sincerely yours,
Shiqi Tian
